# Nutritional and Inflammatory Markers Associated with SARS-CoV-2 Infection in the Elderly

**DOI:** 10.3390/ijms25147749

**Published:** 2024-07-15

**Authors:** João Ismael Budelon Gonçalves, Fernanda Muller Lermen, Júlia Budelon Gonçalves, Gabriele Zanirati, Denise Cantarelli Machado, Helena Morsch Marques, Helena Scartassini Erwig, Bruno Maestri Becker, Fernanda Wagner, Marina Ottmann Boff, Murilo Gomes Rocha, Jaderson Costa Da Costa, e Daniel Marinowic

**Affiliations:** 1Brain Institute of Rio Grande do Sul (BraIns), Pontifical Catholic University of Rio Grande do Sul (PUCRS), Porto Alegre 90610-000, RS, Brazil; joao.goncalves@edu.pucrs.br (J.I.B.G.); fernanda.lermen@edu.pucrs.br (F.M.L.); julia.goncalves@acad.pucrs.br (J.B.G.); gabriele.zanirati@pucrs.br (G.Z.); dcm@pucrs.br (D.C.M.); helena.morsch@edu.pucrs.br (H.M.M.); helena.scartassini@edu.pucrs.br (H.S.E.); bruno.maestri@edu.pucrs.br (B.M.B.); fernanda.wagner002@edu.pucrs.br (F.W.); ottmann.marina@edu.pucrs.br (M.O.B.); murilo.gomes@edu.pucrs.br (M.G.R.); jcc@pucrs.br (J.C.D.C.); 2Graduate Program in Biomedical Gerontology, School of Medicine, Pontifical Catholic University of Rio Grande do Sul (PUCRS), Porto Alegre 90619-900, RS, Brazil

**Keywords:** SARS-CoV-2, COVID-19, nutritional status, inflammation, elderly

## Abstract

The COVID-19 pandemic, caused by the SARS-CoV-2 virus, has posed unprecedented challenges to global health systems, particularly among vulnerable populations such as the elderly. Understanding the interplay between anthropometric markers, molecular profiles, and disease severity is crucial for effective clinical management and intervention strategies. We conducted a cohort study comprising 43 elderly COVID-19 patients admitted to São Lucas Hospital, PUCRS, Brazil. Anthropometric measurements, including calf circumference (CC) and abdominal circumference (AC), were assessed alongside molecular analyses of peripheral blood samples obtained within 48 h of hospital admission. Sociodemographic data were collected from electronic medical records for comprehensive analysis. Our findings revealed a possible relationship between overweight status, increased abdominal adiposity, and prolonged hospitalization duration, alongside heightened disease severity. We also found no significant correlations between BMI, vitamin D levels, and clinical outcomes. Elevated oxygen requirements were observed in both normal and overweight individuals, with the latter necessitating prolonged oxygen therapy. Molecular analyses revealed changes in the inflammatory profile regarding the outcome of the patients. Our study highlights the critical importance of both anthropometric and molecular markers in predicting disease severity and clinical outcomes in elderly individuals with COVID-19.

## 1. Introduction

The COVID-19 pandemic, caused by the novel coronavirus SARS-CoV-2, has presented unprecedented challenges to global health systems and highlighted the urgent need for comprehensive strategies to mitigate its impact [1,2]. Among the various factors influencing COVID-19 outcomes, the role of nutritional status has emerged as a critical yet underexplored determinant of disease severity and progression [3,4]. Nutritional factors are known to significantly influence immune system function, affecting both the body’s defense mechanisms and its response to inflammatory processes [5].

There is a significant relationship between disease severity and immune markers. It has been suggested that during the response to SARS-CoV-2, immune dysregulation and high levels of pro-inflammatory cytokines could be the main cause of tissue damage (https://doi.org/10.1016/j.ebiom.2020.102833 (accessed on 10 July 2024)).

Malnutrition, encompassing both undernutrition and overnutrition, has been associated with impaired immune function, leading to increased susceptibility to infections and a propensity for more severe disease courses [5,6]. In the context of COVID-19, patients with poor nutritional status may exhibit exacerbated inflammatory responses, commonly referred to as cytokine storms, which are linked to severe complications such as acute respiratory distress syndrome (ARDS) and multi-organ failure [7,8,9]. Conversely, adequate intake of specific nutrients, such as vitamins C and D, zinc, and omega-3 fatty acids, has been suggested to modulate immune responses and potentially reduce the severity of infections [10,11,12]. Nutritional status plays a crucial role in maintaining the immune system. Evidence shows that nutrition is one of the most potent extrinsic factors for preventing, modulating, and even reversing immunosenescent states (https://doi.org/10.1590/S1415-52732005000300009 (accessed on 10 July 2024))

Despite these associations, the direct correlation between specific nutritional factors and COVID-19 outcomes remains insufficiently characterized. Most existing studies have focused on general health outcomes related to nutrition or the impact of COVID-19 on the nutritional status, without delving into the mechanistic links between specific nutrients and the inflammatory markers that play pivotal roles in COVID-19 severity. Furthermore, the interplay between nutritional status and other demographic and health-related factors, such as age, comorbidities, and socioeconomic status, complicates the interpretation of existing data.

This study aims to fill some of these gaps by systematically analyzing the nutritional profiles of COVID-19 patients and correlating these profiles with both disease severity and levels of key inflammatory markers. By integrating clinical data with detailed dietary information, this research aimed to help elucidate the potential of nutritional interventions as part of a holistic approach to managing COVID-19, ultimately contributing to more effective treatment protocols and improved patient outcomes.

## 2. Results

Out of the 43 participants, 70% (n = 30) were women, while 30% (n = 13) were men. The average age of the participants was 71.6 ± 8.12 years. As age increased, there was a gradual decrease in representation, with most hospitalized elderly patients fell within the age range of 60–69 years (44.2%), followed by those aged 70–79 years (37.2%). A smaller proportion of patients were aged 80 years or older (18.6%). Most participants were white (90.7%) and married (48.8%) (Table 1).

Table 2 provides an overview of the symptoms and signs observed at admission among elderly patients hospitalized at São Lucas Hospital (HSL) of PUCRS in Porto Alegre, diagnosed with COVID-19, during the period between January and April 2021. The data are categorized based on the patients’ BMI groups, distinguishing between those with normal BMI and those classified as overweight.

Upon admission, none of the patients reported symptoms of a runny nose or nausea. However, approximately 11.6% of patients presented with diarrhea, with no significant variance observed between the normal-BMI and overweight groups (*p* = 0.345). Other symptoms such as body aches, sore throat, headache, fever, loss of appetite, loss of taste, and loss of smell were reported by varying proportions of patients, with no notable differences between the two BMI groups (Table 2).

Fatigue was reported by 25.6% of patients, while cough was the most prevalent symptom, affecting 41.9% of the elderly patients upon admission. Vomiting was reported by a small proportion (2.3%) of patients, with one patient from the normal BMI group experiencing this symptom. Notably, most patients (58.1%) presented with oxygen saturation levels below 95%, underscoring the severity of respiratory compromise among hospitalized elderly patients with COVID-19, irrespective of their BMI status (Table 2). In summary, there were no significant disparities in the prevalence of COVID-19 symptoms and signs at admission between elderly patients with normal BMI and those categorized as overweight.

In terms of clinical characteristics, hypertension, diabetes, and cardiovascular disease were prevalent in 74.4%, 37.2%, and 34.9% of our patients, respectively. All participants used some type of medication, with 83.8% using continuous medication and 46.5% using five or more medications a day (Table 3).

Regarding body mass index (BMI), the data reveals that 41.9% of patients exhibited a normal BMI, while 48.8% were classified as overweight. Conversely, only a small percentage (9.3%) fell into the underweight category, indicating a predominance of patients with either normal or elevated BMI levels (Table 4).

In terms of calf circumference (CC), a larger portion (65.1%) of elderly patients hospitalized with COVID-19 displayed adequate calf circumference, whereas 34.9% exhibited diminished calf circumference (Table 4). This suggests that a significant proportion may be experiencing muscle wasting or malnutrition, potentially indicating underlying health issues. Regarding waist circumference (WC), most patients (69.8%) presented abdominal adiposity, characterized by increased waist circumference, indicating a prevalent occurrence of central obesity among the hospitalized elderly population.

Moreover, the data on vitamin D status reveals a concerning prevalence of deficiency among the patients, with 86.0% classified as deficient. In contrast, only 11.6% had desirable levels of vitamin D, and a small fraction of 2.4% had optimal levels (see Table 4). This highlights a significant nutritional deficiency among elderly patients hospitalized with COVID-19 during the specified period.

When analyzing the association between anthropometric and severity-related parameters with the nutritional status of elderly patients, there was no statistically significant difference between the normal and overweight groups related to age (*p* = 0.77118) (Table 5).

Regarding oxygen saturation (O^2^Sat) levels, there was no significant difference observed between the normal and overweight groups (*p* = 0.87073) (Table 5), indicating that oxygen saturation levels were comparable regardless of nutritional status. Among these elderly individuals with COVID-19, it is noted that the overweight and higher abdominal fat group demonstrates an association with longer hospitalization times (though not statistically significant), increased duration of oxygen use, and more-severe disease compared to the group with normal nutritional status. (Table 5). Finally, vitamin D levels also showed no significant difference between the normal and overweight groups (*p* = 0.3349) (Table 5), indicating comparable vitamin D status regardless of nutritional status.

We conducted correlation analyses to examine the relationships between BMI, vitamin D levels, and clinically relevant outcomes in the hospitalized elderly patients. The outcomes of interest included ICU length of stay, O2 saturation, and the length of O2 supplementation.

In summary, our correlation analyses did not find significant associations between BMI or vitamin D levels and the clinical outcomes of ICU length of stay, O2 saturation, and the length of O2 supplementation in the studied cohort of hospitalized elderly patients with COVID-19 (Figure 1A–F). The Spearman’s correlation coefficients for all analyses were close to zero, and none of the *p*-values indicated statistical significance.

When analyzing the outcomes from the patients, while most patients required supplemental oxygen (86%) and were discharged from the hospital (86%), there were no significant differences in these outcomes between patients with normal and overweight BMI (Table 6). Similarly, there were no significant differences in the rates of ICU admission or mechanical ventilation between the two groups. However, mortality appeared to be higher among patients with normal BMI (83.3%) compared to those who were overweight (16.7%) (Table 6). These findings suggest that while BMI may not significantly influence the need for supplemental oxygen, ICU admission, or mechanical ventilation among elderly COVID-19 patients, it could potentially impact mortality outcomes.

Following the preceding steps, we conducted gene expression analysis using RT-qPCR to examine the levels of inflammation-associated and anti-viral response genes. This analysis was performed on samples obtained from all participants enrolled in the study. Subsequently, we stratified the samples based on the respective groups under investigation (Figure 2A–H).

In patients with higher BMI, there is a pronounced upregulation of inflammatory markers such as TNF-α, which shows a fold change of 33.2; and MAPK14, with a fold change of 6.5 (Figure 2A). Additionally, Caspase-3, involved in apoptosis, is significantly elevated with a fold change of 7. These increases suggest that obesity is linked to a heightened inflammatory and apoptotic response during COVID-19. Concurrently, there is a notable downregulation of interferons IFN-α and IFN-β, with fold changes of −5 and −3.3, respectively, indicating a potential impairment in the antiviral defense mechanisms of obese patients. Other genes, such as VDR, IL-6, and NF-κB, also show moderate changes with fold changes of 2.5, 3.4, and 3.2, respectively. However, genes like MX (−1.5-fold), OAS (0.7-fold), and IFN-y (2-fold) exhibit relatively minor changes, indicating that not all immune pathways are equally affected by obesity.

In the context of vitamin D deficiency, there is a marked increase in IL-6 and caspase-3 expression, with fold changes of 6.5 and 8.4, respectively (Figure 2B). These findings highlight an inflammatory and apoptotic milieu in patients with low vitamin D levels. Furthermore, these patients exhibit decreased levels of IFN-α and IFN-β, with fold changes of −4.7 and −4.3, respectively, suggesting that vitamin D deficiency may contribute to a weakened antiviral response, compounding the severity of the disease. Moderate increases are observed in CXCL10 (4.2-fold) and IL-1β (6-fold), while genes such as OAS (2.7-fold), IFN-y (2.1-fold), and VDR (2.1-fold) show smaller yet noticeable changes. Meanwhile, genes like MX (−1-fold) and NF-κB (1.4-fold) change minimally, indicating a selective impact of vitamin D deficiency on gene expression.

Patients requiring supplementary oxygen also display significant gene expression changes, particularly in IL-6 and CXCL10, which show fold changes of 5.8 and 6.3 (Figure 2C). This indicates an inflammatory response and enhanced chemotaxis. Lesser changes are seen in NF-κB (1.8-fold), VDR (2.1-fold), and IL-1β (2-fold), while IFN-y (4.1-fold) show moderate increases. Notably, antiviral MX (−3.1-fold) and OAS (−3.1-fold) are downregulated. When examining patients who needed supplementary oxygen for more than seven days, there is an even more pronounced increase in pro-inflammatory cytokines TNF-α and IL-1β, with fold changes of 18.5 and 8, respectively (Figure 2D). MCP-1, a chemokine, is also notably elevated with a fold change of 8.7. These findings underscore the intense inflammatory state in patients requiring prolonged oxygen therapy.

ICU admission is associated with increased expression of CXCL10, showing a fold change of 6.7, suggesting a robust chemotactic response in these patients (Figure 2E). However, there is a significant decrease in IFN-β levels, with a fold change of −6.1, pointing to a potential compromise in the antiviral response. Other genes such as TNF-α (4.3-fold), caspase-3 (4.1-fold), MCP-1 (4.2-fold), and IFN-y (5-fold) also show elevated levels. Lesser changes are seen in OAS (−2.3-fold) and IL-1β (3-fold), while genes like MX (1.2-fold) and VDR (0.8-fold) have minimal alterations. For patients staying in the ICU for more than seven days, there are substantial increases in TNF-α and MAPK14, with fold changes of 26.2 and 7.1, respectively (Figure 2F). This further indicates a severe inflammatory response, while the decreased IFN-β levels (−6.1-fold) continue to reflect impaired antiviral defenses. Other notable increases include IL-1β (6.7-fold), caspase-3 (4.2-fold), and MCP-1 (5.8-fold). Moderate changes are observed in VDR (2.1-fold).

Patients with oxygen saturation levels below 95% exhibit increased expression of IL-8 and IL-1β, with fold changes of 7.6 and 8.8, respectively, indicating significant inflammation (Figure 2G). The reduction in IFN-β, with a fold change of −4.1, again points to weakened antiviral mechanisms in these patients. Caspase-3 (3.4-fold) shows a moderate increase, while VDR (1.7-fold) and TNF-α (0.2-fold) are less affected.

Furthermore, the most severe outcome, death, is linked to dramatic changes in gene expression. There is a notable increase in MAPK14, IL-17, and IL-4, with fold changes of 13.2, 7.2, and 6.1, respectively, highlighting a potent inflammatory response and activation of Th2/Th17 pathways (Figure 2H). The reduced levels of IFN-β, with a fold change of −4.2, further emphasize the impaired antiviral response in these patients. Other notable increases include IL-6 (4.3-fold), IL-1β (3-fold), and TNF-α (3.5-fold), further highlighting the presence of an inflammatory response. Moderate changes are seen in VDR (1.6-fold) and IFN-γ (3.7-fold) while genes such as MX (−1.8-fold) and OAS (−2.3-fold) show less-significant variations.

To further confirm the association we observed between BMI, vitamin D levels, and inflammatory markers, we analyzed the serum levels of selected pro- and anti-inflammatory cytokines. When we examined the cytokines based on patients’ BMI, we found significant elevations in the inflammatory cytokines TNF-α (Figure 3A) (*p* = 0.0029), IL-1β (Figure 3B) (*p* = 0.0363), and IL-6 (Figure 3C) (*p* = 0.0112); whereas IL-17 (Figure 3D) and IL-10 (Figure 3E) showed no significant changes. In contrast, analysis of cytokines based on patients’ vitamin D levels revealed no significant differences in TNF-α levels between those with normal and deficient vitamin D levels (Figure 3F). However, IL-1β (Figure 3G) (*p* = 0.0054) and IL-6 (Figure 3H) (*p* ≤ 0.0001) were significantly elevated in patients with low vitamin D levels, while IL-17 (Figure 3I) and IL-10 (Figure 3J) remained unchanged. These analyses suggest that overweight and vitamin D deficient COVID-19 patients have significantly higher serum inflammatory cytokine levels, indicating a potential association between nutritional status and inflammatory response in COVID-19.

## 3. Discussion

The present study aimed to evaluate the interplay between BMI, vitamin D levels, inflammation-associated gene expression, and clinical variables among elderly patients hospitalized due to COVID-19. The multifaceted analysis provided valuable insights into potential associations and their implications for patient outcomes.

Nutritional assessment revealed concerning trends, with a significant proportion of patients classified as overweight, as indicated by increased BMI and waist circumference. Moreover, a considerable number of patients were found to be deficient in vitamin D, highlighting potential nutritional deficiencies among this population. These findings underscore the importance of nutritional assessment and interventions in managing COVID-19 outcomes among elderly patients.

Analysis of clinical characteristics revealed a high prevalence of comorbidities, particularly hypertension, diabetes, and cardiovascular disease, which are known risk factors for severe COVID-19 outcomes [13]. Additionally, the high proportion of patients on multiple medications reflects the complex medical needs of this population.

Regarding BMI correlations, our findings revealed no associations with clinical variables such as oxygen saturation, vitamin D levels, and the duration of oxygen supplementation. Although statistical significance was not reached, trends suggested that higher BMI values might be linked to slightly lower O2 saturation levels and lower vitamin D levels. It is important to note that the low sample number in our study could be a contributing factor to the lack of significant findings. Larger sample sizes may be needed to detect potential correlations and better understand the relationships between these variables. Therefore, further investigation is warranted to confirm these trends and ascertain any significant impact of BMI on clinical outcomes in this population. Higher BMI has been linked to an increased risk of more-severe COVID-19, including higher rates of hospitalization and mortality [14]. This is thought to be due to the impact of obesity on respiratory function and inflammation [15].

Similarly, correlations between vitamin D levels and clinical variables showed no associations without statistical significance. While vitamin D deficiency appeared to be linked to increased inflammation, as evidenced by elevated IL-6 expression, the lack of significant relationships with other clinical outcomes implies that vitamin D levels may not independently influence patient outcomes in elderly COVID-19 patients. Recent studies suggest that vitamin D deficiency may be associated with an increased risk of severe COVID-19 outcomes [16,17,18]. The relatively small sample size of our study, in contrast to the multicenter cohort studies that report similar associations, could explain the discrepancies observed between our findings and those of other reports.

The findings of this study underscore the significant impact of BMI and vitamin D levels on the inflammatory response in COVID-19 patients, evidenced by both gene expression and serum cytokine analyses. Our data reveal pronounced upregulation of inflammatory markers in patients with higher BMI, suggesting that obesity is linked to an intensified inflammatory and apoptotic response during COVID-19. Specifically, the significant increases in TNF-α, IL-1β, and IL-6, as well as the elevated expression of genes such as MAPK14 and caspase-3, highlight the heightened inflammatory state in obese patients. TNF-α, a pro-inflammatory cytokine primarily secreted by monocytes and macrophages during acute inflammatory periods, plays a crucial role in initiating cell signaling processes that can lead to necrosis and apoptosis [19]. Our study revealed differences in patients requiring prolonged oxygen support; they exhibited increased relative expression of the TNF-α gene compared to those requiring support for shorter durations. This finding is supported by previous research indicating elevated TNF-α levels in patients with severe SARS-CoV-2 infection [20,21]. Furthermore, abnormal elevations in TNF-α production have been associated with poor prognosis in viral infections and adverse outcomes [22,23].

The concurrent downregulation of interferons IFN-α and IFN-β in obese patients suggests a compromised antiviral defense mechanism, which may further exacerbate the severity of COVID-19 in these individuals. The observed changes in other genes like VDR, IL-6, and NF-κB, although moderate, reinforce the idea that obesity affects multiple facets of the immune response. The minimal changes in genes such as MX, OAS, and IFN-γ indicate that not all immune pathways are equally influenced by obesity, pointing to a complex interaction between obesity and immune function.

Vitamin D, categorized as a steroid hormone, is synthesized by the human skin upon exposure to sunlight [24]. This hormone plays a crucial role in modulating both innate and adaptive immunity [25], and its deficiency has been associated with increased susceptibility to infections [24,26]. In patients with vitamin D deficiency, the increases in IL-6 and caspase-3 expression, alongside elevated levels of IL-1β and CXCL10, emphasize a pronounced inflammatory and apoptotic milieu. The decreases in IFN-α and IFN-β levels in these patients suggest a weakened antiviral response, which could contribute to the increased severity of COVID-19 observed in vitamin D deficient individuals. These findings align with previous studies that have linked vitamin D deficiency to worse outcomes in infectious diseases due to its role in modulating the immune response [27,28].

The stratification of patients based on their need for supplementary oxygen further highlights the severity of inflammation in those requiring prolonged oxygen therapy. Elevated levels of pro-inflammatory cytokines such as TNF-α and IL-1β, along with increased expression of chemokines like MCP-1 and CXCL10, suggest an intense inflammatory state that may necessitate extended respiratory support. The downregulation of antiviral genes such as MX and OAS in these patients underscores the potential compromise in antiviral defenses, compounding the challenges in managing severe COVID-19 cases.

Patients admitted to the ICU or with low oxygen saturation levels exhibited upregulation of inflammatory markers and downregulation of key antiviral genes, reflecting a severe and dysregulated immune response. The notable increases in TNF-α, IL-1β, and MAPK14 in these patients further support the link between heightened inflammation and poor clinical outcomes. The observed decrease in IFN-β levels consistently across various severe patient groups suggests that impaired antiviral responses are a common feature of severe COVID-19, irrespective of BMI or vitamin D status [29].

Our study has several limitations. First, the relatively small sample size may constrain the generalizability of our findings to broader populations, underscoring the need for larger, more diverse cohorts. Second, the cross-sectional design of the study inherently limits our ability to establish causal relationships between gene expression profiles, clinical variables, and outcomes. Without longitudinal data, we cannot determine whether the observed changes in gene expression are a cause or consequence of the clinical conditions studied. Longitudinal investigations are essential to understand the temporal dynamics and potential causality of these associations, providing more robust insights into the mechanisms driving inflammation and disease progression. Moreover, the study may not have fully accounted for confounding variables, such as medication use and lifestyle factors, which could influence both gene expression patterns and clinical outcomes. Lastly, while associations between gene expression levels and clinical variables were identified, the biological significance and clinical implications of these findings remain uncertain. Further elucidation through functional studies and validation in independent cohorts is required. Addressing these limitations in future research will be crucial for advancing our understanding of the molecular mechanisms driving inflammation and clinical outcomes in elderly COVID-19 patients. Future research endeavors should focus on addressing these limitations to enhance the validity and generalizability of the findings and provide more definitive insights into the molecular underpinnings of inflammation and clinical outcomes in COVID-19 patients.

Overall, our study demonstrates that both high BMI and low vitamin D levels are associated with increased inflammatory responses and compromised antiviral defenses in COVID-19 patients. These findings suggest that addressing nutritional status, particularly through weight management and vitamin D supplementation, could be crucial in reducing the severity of COVID-19. Future research should explore the mechanisms underlying these associations and evaluate the potential benefits of targeted nutritional and anti-inflammatory interventions in improving patient outcomes. Understanding the interplay between nutritional status and immune response will be key to developing effective strategies to combat COVID-19 and similar infectious diseases.

## 4. Materials and Methods

### 4.1. Ethical Aspects

This study was approved by the Committee of Research Ethics of the Pontifical Catholic University of Rio Grande do Sul (CAAE: 40291320.1.0000.5336 approval number: 4.441.775). The free and informed consent form was obtained from all study participants in accordance with Resolution No. 466/12 of the Brazilian National Health Council.

### 4.2. Population and Sample

This study enrolled forty-three participants aged 60 years or older, irrespective of gender, who were admitted to the inpatient unit of São Lucas Hospital of PUCRS, Brazil, with confirmed COVID-19 diagnosis and provided their consent to participate. This was a convenience sample where all hospitalized patients were invited to participate in the study. Patients without physical conditions to be evaluated, insufficient peripheral blood collection, and incomplete medical records were excluded from the study. Data collection involved accessing the medical records of the participants, covering the period from January to April 2021.

### 4.3. Anthropometric Evaluation

Anthropometric evaluations involved direct measurements of weight and height recorded in the medical records. The body mass index (BMI) was calculated by dividing weight (in kilograms) by the square of height (in meters) and categorized based on Lipschitz cutoff points and the Nutrition Screening Initiative guidelines. Additionally, calf circumference (CC) and abdominal circumference (AC) were assessed using a flexible and inelastic tape (Avanutri^®^ brand, model AVA-04). For CC measurement, the tape was positioned at the widest part of the calf, perpendicular to its length, and the circumference was recorded without exerting pressure or leaving slack. Multiple measurements were taken above and below this point to ensure accuracy, with the largest measurement recorded. Classification followed cutoff points outlined in the BRASPEN Guideline on nutritional therapy in aging [30]. AC measurement involved locating the midpoint between the last rib and the iliac crest, where the circumference was measured without pressure. The recorded value was where the tape coincided with zero. Classification was based on cutoff points described in the I Brazilian Guideline for the diagnosis and treatment of metabolic syndrome [31].

### 4.4. Collection of Demographic Data

By accessing participants’ electronic medical records, we collected sociodemographic data encompassing variables such as sex, age, marital status, race, symptoms, underlying medical history, and previous continuous medication use. This information was then recorded in a spreadsheet for subsequent analysis.

### 4.5. Peripheral Blood Collection

For molecular and biochemical analyses, 4 mL of peripheral blood was drawn into a collection tube containing EDTA anticoagulant within 48 h of hospitalization. Following collection, the material underwent centrifugation at 400× *g* for 5 min to isolate the plasma fraction, which was then stored at −80 °C. The fraction containing blood nucleated cells, located between the plasma and red blood cells, was collected. Subsequently, 500 μL of red blood cell lysis buffer was added to this fraction and incubated for 5 min at room temperature. Afterward, the mixture was centrifuged at 12,000× *g* for 5 min, and the separated cells were resuspended in 200 μL of RNA Later Stabilization solution (Invitrogen, Waltham, MA, USA) before being stored at −80 °C.

### 4.6. Vitamin D Quantification

Vitamin D levels in the blood were quantified using the 25(OH) Vitamin D ELISA kit (Abcam, Cambridge, UK) following the manufacturer’s instructions. The assay utilized monoclonal antibodies specific to vitamin D, enabling accurate quantification of 25-hydroxyvitamin D [25(OH)D], the primary circulating form of vitamin D.

### 4.7. RNA Extraction and cDNA Synthesis

RNA extraction from peripheral blood mononuclear cells (PBMCs), collected and stored as described above, was performed using SV-Total RNA kit (Promega, Madison, WI, USA) according to the manufacturer. The extracted RNA was diluted in 100 μL of nuclease-free water and converted to cDNA using the Go Script Oligo dT (Promega) kit. cDNA was quantified in NanoDrop (Thermo Fisher Scientific, Waltham, MA, USA) and the concentration required for qRT-PCR analysis was calculated.

### 4.8. Molecular Analysis by qRT-PCR

Real-time quantitative PCR was performed using the PowerUp SYBR Green (Thermo Fisher Scientific) master mix on the StepOne Plus (Thermo Fisher Scientific) equipment. The samples were amplified with an initial value of 100 ng of single-stranded DNA (ssDNA) for each sample. The genes analyzed and the sequence of primers are described in Appendix A. The gene *Actb* was used as the reference for normalization. The relative expression levels of the target genes were calculated using the 2^−ΔΔCt^ method, where ΔCt = Ct(target) − Ct(reference) and ΔΔCt = ΔCt(sample) − ΔCt(control).

### 4.9. Luminex Immunoassay

The levels of TNF-α, IL-1β, IL-6, IL-10, and IL-17 in the serum samples were quantified using a multiplex bead-based Luminex assay (MilliporeSigma, Burlington, MA, USA). The assay was performed according to the manufacturer’s instructions. Reagents, including standards and controls, were prepared as per the manufacturer’s instructions. Serum samples were thawed on ice and diluted appropriately in assay buffer. A 96-well microplate was pre-wetted with assay buffer. Next, 50 µL of standard, control, or diluted serum sample was added to each well, followed by 50 µL of the mixed antibody-immobilized beads. The plate was covered and incubated on a plate shaker at 800 rpm for 2 h at room temperature in the dark. The plate was then washed three times with 200 µL of wash buffer using a magnetic plate washer to remove unbound substances. After washing, 50 µL of biotinylated detection antibody was added to each well and incubated on a plate shaker at 800 rpm for 1 h at room temperature in the dark. The plate was washed again three times with 200 µL of wash buffer, followed by the addition of 50 µL of streptavidin-phycoerythrin to each well. The plate was incubated on a plate shaker at 800 rpm for 30 min at room temperature in the dark. Following the final wash, the beads were resuspended in 100 µL of wash buffer, and the plate was read using a Luminex MAGPIX instrument (Luminex Corporation, Austin, TX, USA). Data were acquired using xPONENT 4.3 software.

### 4.10. Statistical Analysis

Statistical analysis was performed using the software Prism9 (Graphpad Software Inc., La Jolla, CA, USA). Associations between categorical variables were tested using Pearson’s Chi-square test and, in specific cases, the Chi-square test for linear trend regression. Differences between specific points were determined via Student’s *t*-test for normal distribution and Mann–Whitney U test for non-normal distributions. To assess the strength and direction of the association between continuous variables (BMI and vitamin D levels), Spearman’s rank correlation test was employed. Linear regression was used to explore the impact of BMI and vitamin D levels on continuous outcomes related to COVID-19 severity.

## Figures and Tables

**Figure 1 ijms-25-07749-f001:**
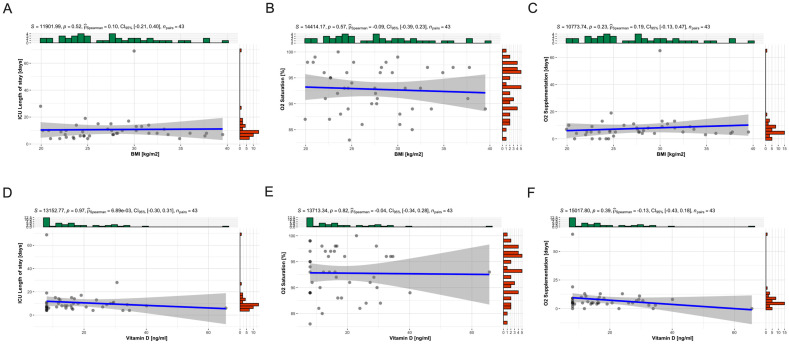
Correlation analyses between BMI and vitamin D and clinically relevant outcomes. Scatter plots of BMI and vitamin D results according to outcomes of hospitalized elderly patients at São Lucas Hospital (HSL) of PUCRS in Porto Alegre, diagnosed with COVID-19. We performed correlation analyses between: (**A**) BMI and ICU length of stay; (**B**) BMI and O_2_ saturation; (**C**) BMI and length of O_2_ supplementation; (**D**) vitamin D and ICU length of stay; (**E**) vitamin D and O_2_ saturation; (**F**) vitamin D and length of O_2_ supplementation. The analysis was performed using Spearman’s rank correlation test. Each plot shows individual data points with a fitted regression line (blue) and 95% confidence interval (shaded area).

**Figure 2 ijms-25-07749-f002:**
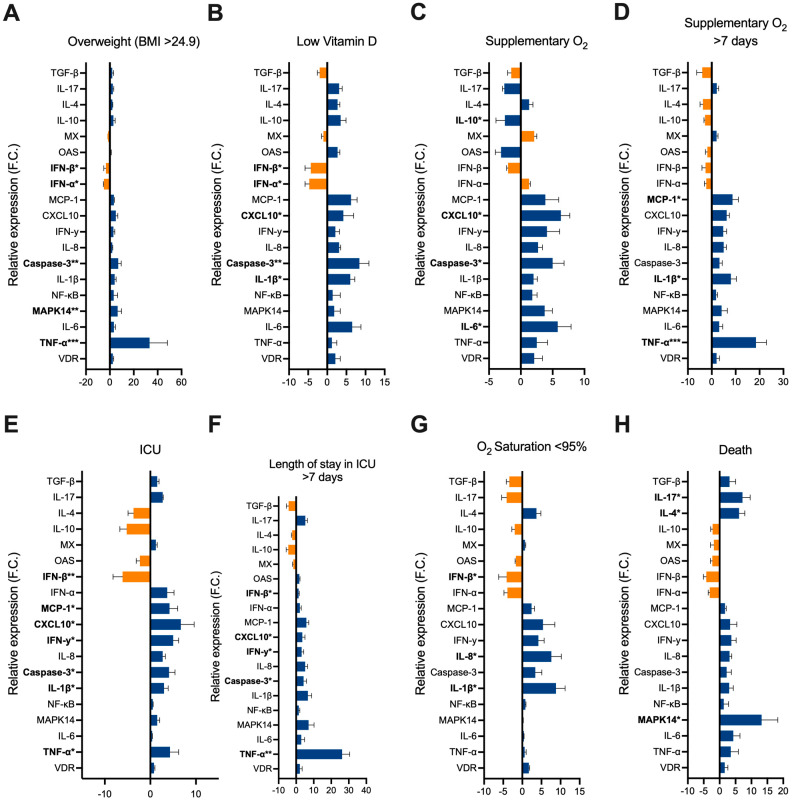
Representation of the results of relative expression of the analyzed genes. Expression of inflammatory markers were analyzed in stratified groups of patients, based on clinically relevant characteristics, including (**A**) BMI, (**B**) vitamin D levels, (**C**) use of supplementary O^2^, (**D**) use of supplementary O^2^ > 7 days, (**E**) O_2_ saturation; (**F**) ICU admission, (**G**) length of stay in ICU > 7 days, and (**H**) death. The values are presented through the calculation of relative expression 2^−ΔΔ^ CT and normalized using β-actin as endogenous control. The data presented are from three independent experiments, and asterisks on gene’s names indicate statistical significance as follows: * 0.01 < *p* < 0.05, ** 0.001 < *p* < 0.01, *** *p* < 0.001. The Mann–Whitney U test was used to determine statistical significance. Error bars represent SEM.

**Figure 3 ijms-25-07749-f003:**
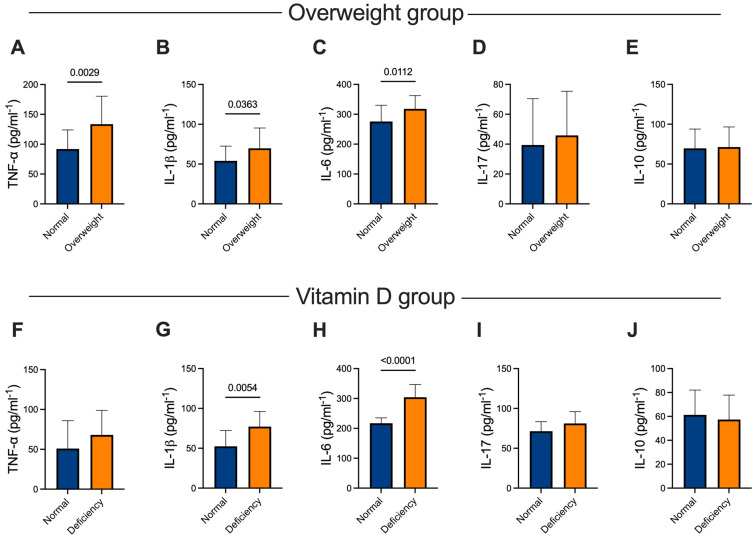
Serum values of pro-inflammatory and anti-inflammatory cytokines. Patients were segregated based on their BMI status and vitamin D levels. Systemic levels of IL-1β, IL-6, IL-17, and IL-10 were measured via Luminex immunoassay. Statistical significance was tested by the Student’s *t*-test.

**Table 1 ijms-25-07749-t001:** Sociodemographic characteristics of elderly patients hospitalized at São Lucas Hospital (HSL) of PUCRS in Porto Alegre, diagnosed with COVID-19, between January and April 2021.

Variables		N	(%)
Age	60–69	19	(44.2)
	70–79	16	(37.2)
	≥80	8	(18.6)
Gender	Female	30	(69.8)
	Male	13	(30.2)
Marital Status	Single	11	(25.6)
	Married	21	(48.8)
	Divorced	4	(9.3)
	Widowed	7	(16.3)
Race	White	39	(90.7)
	Black	4	(9.3)
TOTAL		43	100%

Legend: N = number.

**Table 2 ijms-25-07749-t002:** Nutritional characteristics of elderly patients hospitalized at São Lucas Hospital (HSL) of PUCRS in Porto Alegre, diagnosed with COVID-19, between January and April 2021.

Variables		N	(%)
BMI	Underweight (<18.5)	4	(9.3)
	Normal (18.5–24.9)	18	(41.9)
	Overweight (≥25)	21	(48.8)
CC	Adequate (≥34 cm)	28	(65.1)
	Diminished (≥34 cm)	15	(34.9)
WC	Adiposity (≥88 cm (women)/102 cm (men))	30	(69.8)
	Adequate (<88 cm (women)/102 cm (men))	13	(30.2)
Vitamin D	Deficient (<20 ng/mL)	37	(86.0)
	Desirable (20–29 ng/mL)	5	(11.6)
	Optimal (≥30 ng/mL)	1	(2.4)
TOTAL		43	100%

Legend: N = number; BMI = body mass index; CC = calf circumference; WC = waist circumference.

**Table 3 ijms-25-07749-t003:** Association of anthropometric and severity-related parameters with nutritional status of elderly patients hospitalized at São Lucas Hospital (HSL) of PUCRS in Porto Alegre, diagnosed with COVID-19, between January and April 2021.

	Normal	Overweight	
	Mean ± Std Dev	Mean ± Std Dev	*p*-Value
Age	71.3 ± 8.51	72.0 ± 7.73	0.77118
Weight	65.2 ± 10.76	82.7 ± 11.82	0.00001
WC	93.0 ± 12.29	110.6 ± 11.30	0.00002
CC	32.5 ± 3.05	36.7 ± 3.82	0.00016
O^2^Sat	92.7 ± 4.70	92.9 ± 4.19	0.87073
Hospitalization days	9.0 ± 5.79	12.4 ± 13.35	0.28378
Days with O^2^	5.6 ± 4.86	9.7 ± 13.00	0.18188
Vitamin D	20.1 ± 13.60	16.6 ± 9.21	0.3349

Legend: WC = waist circumference; CC = calf circumference; O^2^Sat = oxygen saturation.

**Table 4 ijms-25-07749-t004:** Symptoms and signs at admission of elderly patients hospitalized at São Lucas Hospital (HSL) of PUCRS in Porto Alegre, diagnosed with COVID-19, between January and April 2021.

	Total	BMI Group		
Characteristics	n = 43	Normal	Overweight	*p*-Value
		n = 22	n = 21	
Runny Nose	0	0	0	
Diarrhea	5 (11.6%)	4 (80%)	1 (20%)	0.344866705
Body Aches	3 (7%)	2 (66.7%)	1 (33.3%)	1
Sore Throat	1 (2.3%)	0	1 (100%)	0.488372093
Headache	3 (7%)	0	3 (100%)	0.107770845
Fever	13 (30.2%)	5 (38.5%)	8 (61.5%)	0.33188888
Nausea	0	0	0	
Loss of Appetite	4 (9.3%)	2 (50%)	2 (50%)	1
Loss of Taste	3 (7%)	0	3 (100%)	0.107770845
Loss of Smell	3 (7%)	0	3 (100%)	0.107770845
Fatigue	11 (25.6%)	8 (72.7%)	3 (27.3%)	0.097196099
Cough	18 (41.9%)	8 (44.4%)	10 (55.6%)	0.454554972
Vomiting	1 (2.3%)	1 (4.6%)	0	1
O_2_Sat < 95%	25 (58.1%)	13 (52%)	12 (48%)	0.897013982

Note: The *p*-values refer to the Chi-square test. Legend: O_2_Sat = oxygen saturation.

**Table 5 ijms-25-07749-t005:** Clinical characteristics of elderly patients hospitalized at São Lucas Hospital (HSL) of PUCRS in Porto Alegre, diagnosed with COVID-19, between January and April 2021.

	Total	BMI Group		
Characteristics	n = 43	Normal	Overweight	*p*-Value
		n = 22	n = 21	
Diabetes	16 (37.2%)	8 (50%)	8 (50%)	0.906523106
Cardiovascular Disease	15 (34.9%	7 (46.7%)	8 (53.3%)	0.665956184
Cerebrovascular Disease	0	0	0	
Pulmonary Disease	1 (2.3%)	0	1 (100%)	0.488372093
Hypertension	32 (74.4%)	15 (46.9%)	17 (53.1%)	0.337363721
Chronic Medication	36 (83.8%)	17 (47.2%)	19 (52.8%)	0.412059863
>5 Medications	20 (46.5%)	9 (45%)	11 (55%)	0.450912971

Note: The *p*-values refer to the Chi-square test.

**Table 6 ijms-25-07749-t006:** Hospital outcomes of elderly patients hospitalized at São Lucas Hospital (HSL) of PUCRS in Porto Alegre, diagnosed with COVID-19, between January and April 2021.

	Total	Grupo de IMC		
Characteristics	n = 43	Normal	Overweight	*p*-Value
		n = 22	n = 21	
Supplemental O^2^	37 (86%)	17 (45.9%)	20 (54.1%)	0.185282953
ICU Admission	6 (13.9%)	3 (50%)	3 (50%)	1
Mechanical Ventilation	5 (11.6%)	3 (60%)	2 (40%)	1
Discharge Outcome	37 (86%)	17 (45.9%)	20 (54.1%)	0.185282953
Death Outcome	6 (14%)	5 (83.3%)	1 (16.7%)	

Note: The *p*-values refer to the Chi-square test.

## Data Availability

The original contributions presented in the study are included in the article/Appendix A. Further inquiries can be directed to the corresponding author.

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
