# Peer review of "Nutritional and Inflammatory Markers Associated with SARS-CoV-2 Infection in the Elderly"

_ijms, 2024, doi:10.3390/ijms25147749_

Round 1

Reviewer 1 Report

Comments and Suggestions for Authors

Overall, the article is well-written and focuses on essential health issues. There are specific gaps that authors may fill as a result of my comments:

ABSTRACT

1.       The authors mention that they found a “significant association between overweight status, increased abdominal adiposity, and prolonged hospitalization duration, alongside heightened disease severity.” However, there is no mention of statistical methods used in the study.

INTRODUCTION:

2.       The Introduction section will benefit from more detailed search gaps their research aims to fill. This would help readers understand the justification of the current study and its contribution to existing literature on the subject.

METHODS:

3.       In the section on “Population and Sample,” the reader will benefit from additional details about the sampling methods. Those details must cover the sampling design, its justification, the sample selection method, and the implication of the sampling design on the representability of the population and the generalizability of the findings.

4.       The section “4.3. Anthropometric Evaluation” seems misplaced as it appears before the “Data Collection.” Since data are collected in the “Anthropometric Evaluation” process, it should be wrapped into the data collection section.

5.       The research is based on a small sample size;  a larger sample size with appropriate multivariable regression should have been employed.

6.       The authors used Student’s t-test, Pearson correlation, and Linear Regression. However, they do not justify the appropriateness of these specific methods.

7. The authors’ use of correlation is justifiable due to the small sample size, but correlations are generally inappropriate for establishing associations. The correlation coefficient may provide misleading information by highlighting spurious relationships. Even when strong correlations are found between variables, they do not account for the confounding impact of variables not included in the analysis (through a multivariable analysis).

RESULTS:

8.       The results seem to be written well.

DISCUSSION:

9.       The discussion section is written well.

Comments on the Quality of English Language

Overall, the quality of the English language is acceptable.

Author Response

ABSTRACT

  1. The authors mention that they found a “significant association between overweight status, increased abdominal adiposity, and prolonged hospitalization duration, alongside heightened disease severity.” However, there is no mention of statistical methods used in the study.

We changed this sentence for “possible relationship between overweight status, increased abdominal adiposity, and prolonged hospitalization duration, alongside heightened disease severity”.

INTRODUCTION:

  1. The Introduction section will benefit from more detailed search gaps their research aims to fill. This would help readers understand the justification of the current study and its contribution to existing literature on the subject.

We revised the introduction section to strengthen the Introduction section and provided a clearer rationale for our study. We hope these changes meet your expectations and enhance the overall quality and impact of our manuscript.

METHODS:

  1. In the section on “Population and Sample,” the reader will benefit from additional details about the sampling methods. Those details must cover the sampling design, its justification, the sample selection method, and the implication of the sampling design on the representability of the population and the generalizability of the findings.

We appreciate your suggestion to provide more comprehensive details about our sampling methods. In response, we have expanded the "Population and Sample" section with information about the recruitment of patients and exclusion criteria.

  1. The section “4.3. Anthropometric Evaluation” seems misplaced as it appears before the “Data Collection.” Since data are collected in the “Anthropometric Evaluation” process, it should be wrapped into the data collection section.

Thanks for the observation. We have replaced item 4.4 with "Collection of demographic data”.

  1. The research is based on a small sample size; a larger sample size with appropriate multivariable regression should have been employed.

Thank you for your observation. We understand the importance of analyses such as multivariable regression in population studies like ours. However, we recruited all patients hospitalized at São Lucas Hospital – PUCRS during the pandemic period. This research was conducted during a critical phase of the COVID-19 pandemic, and our access was limited to this hospitalization unit. Consequently, the number of participants was somewhat restricted.

  1. The authors used Student’s t-test, Pearson correlation, and Linear Regression. However, they do not justify the appropriateness of these specific methods.

We used the t-test to compare the means of two groups and Pearson correlation to reveal the relationship between two variables.

We inserted the following sentence in the Statistical Analysis section: Associations between categorical variables were tested using Pearson's Chi-square test and, in specific cases, the Chi-square test for linear trend regression.

  1. The authors’ use of correlation is justifiable due to the small sample size, but correlations are generally inappropriate for establishing associations. The correlation coefficient may provide misleading information by highlighting spurious relationships. Even when strong correlations are found between variables, they do not account for the confounding impact of variables not included in the analysis (through a multivariable analysis).

We used correlation tests but did not find any statistical associations in our analysis. We could remove this analysis from the paper; however, we believe it illustrates an attempt to correlate the obtained data, thereby facilitating the reader's understanding.

RESULTS:

  1. The results seem to be written well.

Thank you very much.

DISCUSSION: 

  1. The discussion section is written well.

Thank you very much.

Reviewer 2 Report

Comments and Suggestions for Authors

The manuscript entitled “Nutritional and inflammatory markers associated with SARS-CoV-2 infection in the elderly” deals with obesity, nutritional and inflammatory status of elderly patients in relation to severity of COVID-19. The hot topic of this manuscript is relevant for understanding the relationship of anthropometric characteristics and potential molecular markers with COVID-19 severity and it is within the scope of the journal.  However, there are some major issues and serious concerns about this manuscript:

- One of the major issues is a very small number of participants. After segregation according to BMI and other criteria, this results in a negligible number of patients within subgroups. The strength of the study is small, even for a pilot study and the lack of statistical significance for some of the differences (obvious in comparison of percentages) could be a consequence of this issue. Any statistical analysis regarding the data presented in Table 4 would result in the lack of statistical significance. On the other hand, the most significant results, according to the authors, stated in the Abstract (lines 22-24), as well as in the Results and Discussion are exaggerated and the major conclusions are not supported by the results. All results related to correlation tests are insignificant and medically and biologically completely irrelevant. Such correlation coefficients are not indicative of any relevant correlation. It is concerning that the authors postulate these results as some of the most significant findings. Therefore, the major part of the Results and Conclusions needs to be corrected. The sentence in lines 173-175 is one of the major examples of the misinterpretation of the results.

- The second major problem is that some of the evaluated variables clearly show deviations from the normal distribution. Therefore, statistical tests used in this research are not suitable for analyzing differences in the status of these variables. This stands not only for Student’s T-test, but also for the applied correlation test.

- There are no statistical parameters describing the results of qPCR-based relative quantification. Without indicators of statistical significance, the relevance of these data presented in Figure 2 cannot be evaluated. Since the number of participants in some of these groups made according to some of severity criteria is less than 10, we can expect that at least some of the relevant “fold changes” are not statistically significant and that this whole section of the results on pages 8 and 9 should be rewritten.

- Data presented in Figure 3 has no matching descriptive section in the Results.

- The order and the content of the tables do not match the text of the Results. Tables have to be reordered. For instance, descriptive results stated for Table 2 match the content of Table 4.

- The second column of Table 2 should incorporate threshold values in parentheses for stated categories.

- Some of the results should not be highlighted, like WC and CC difference between normal and overweight, since this is expected.

- The major limitations need to be elaborated, in accordance to my previous comments.

- Material and Methods section requires additional information. From which cells was the RNA extracted? What was the normalization strategy for qPCR? Which gene was chosen as a reference? How was fold change calculated and which statistical test was applied for assessing the differences between groups?

Comments on the Quality of English Language

Minor editing is needed.

Author Response

One of the major issues is a very small number of participants. After segregation according to BMI and other criteria, this results in a negligible number of patients within subgroups. The strength of the study is small, even for a pilot study and the lack of statistical significance for some of the differences (obvious in comparison of percentages) could be a consequence of this issue. Any statistical analysis regarding the data presented in Table 4 would result in the lack of statistical significance. On the other hand, the most significant results, according to the authors, stated in the Abstract (lines 22-24), as well as in the Results and Discussion are exaggerated and the major conclusions are not supported by the results. All results related to correlation tests are insignificant and medically and biologically completely irrelevant. Such correlation coefficients are not indicative of any relevant correlation. It is concerning that the authors postulate these results as some of the most significant findings. Therefore, the major part of the Results and Conclusions needs to be corrected. The sentence in lines 173-175 is one of the major examples of the misinterpretation of the results.

Answer: We understand that the number of participants is somewhat limited. Segregation based on certain parameters further reduces the number of participants per group. We used this approach to analyze these parameters among participants, as all hospitalized patients who agreed to participate in the study were enrolled without projecting a control group. A control group (without exposure to SARS-CoV-2) during the pandemic became impractical and would have introduced a greater bias than a reduced number of individuals.

We agree with your point regarding the data in Table 4; however, we do not consider it dispensable for the study

Regarding lines 22-24, we have changed them to the following sentence: “possible relationship between overweight status, increased abdominal adiposity, and prolonged hospitalization duration, alongside heightened disease severity”.

We have revised some of the data in the results and discussion sections to present the data in a more fluid and consistent manner with the findings.

Regarding the correlation tests, at no point in the article were the findings presented as correlated. In each presentation of results, we took due care to note the lack of statistical significance of the p-value. We consider this analysis quite illustrative for the reader, but we can consider removing it from the manuscript if it is mandatory.

The second major problem is that some of the evaluated variables clearly show deviations from the normal distribution. Therefore, statistical tests used in this research are not suitable for analyzing differences in the status of these variables. This stands not only for Student’s T-test, but also for the applied correlation test.

Answer: We applied statistical tests according to the distribution of the groups. For the analyses described in Tables 3, 4, 5, and 6, the chi-square test was used. The PCR assays were presented as differences in fold change values relative to baseline 1, shown as gene expression profiles without applying a statistical test for comparison. If mandatory, we can apply statistical analysis to these PCR data. For the luminex assays, the t-test was used due to the parametric distribution of the groups for these results.

There are no statistical parameters describing the results of qPCR-based relative quantification. Without indicators of statistical significance, the relevance of these data presented in Figure 2 cannot be evaluated. Since the number of participants in some of these groups made according to some of severity criteria is less than 10, we can expect that at least some of the relevant “fold changes” are not statistically significant and that this whole section of the results on pages 8 and 9 should be rewritten.

Answer: We chose not to apply a statistical test to the PCR results because the benchmark for fold change is constant (in this case, 1) for all the genes analyzed. For example, as we described in line 202: “In patients with higher BMI, there is a pronounced upregulation of inflammatory markers such as TNF-α, which shows a fold change of 33.2, and MAPK14, with a fold change of 6.5.” For example, the TNF-α gene was expressed 33.2 times more in patients with high BMI compared to patients with low BMI (where the latter is assigned a value of 1). We can apply a statistical test to all these results; however, they may not have the desired impact. The PCR results were described in a manner we consider appropriate, presenting the main fold change values for the genes analyzed between the groups.

Data presented in Figure 3 has no matching descriptive section in the Results.

Answer: We apologize for that. We added a paragraph in the Results section describing the data presented in Figure 3.

The order and the content of the tables do not match the text of the Results. Tables have to be reordered. For instance, descriptive results stated for Table 2 match the content of Table 4.

Answer: We apologize for that. We corrected the order of the Tables.

The second column of Table 2 should incorporate threshold values in parentheses for stated categories.

Answer: We added threshold values for all the categories presented in this Table.

Some of the results should not be highlighted, like WC and CC difference between normal and overweight, since this is expected.

Answer: We removed form text the expected and obvious results to avoid redundancy.

The major limitations need to be elaborated, in accordance to my previous comments.

Answer: We included in the discussion section a more elaborated paragraph highlighting our limitations in the study regarding sample size and statistical significance of the findings.

Material and Methods section requires additional information. From which cells was the RNA extracted? What was the normalization strategy for qPCR? Which gene was chosen as a reference? How was fold change calculated and which statistical test was applied for assessing the differences between groups?

Answer: We added in the Material and Methods section the cells from which RNA was extracted as well as the reference gene used and the method for fold change calculation.

Round 2

Reviewer 1 Report

Comments and Suggestions for Authors

I thank the authors for addressing my comments appropriately. 

Author Response

Review Report (Reviewer 1)

I thank the authors for addressing my comments appropriately

Answer: Thank you very much for you assessment and review

Reviewer 2 Report

Comments and Suggestions for Authors

The authors have made some corrections, however, the improvements are not sufficient for a meaningful increase in the quality of the manuscript. The corrections are mostly intended to account for major flaws in the presentation of the results (Table orders, missing data...) but the most significant issues with the study design and interpretation persist. Stating these issues as a limitation does not change the fact that the results and their interpretation could be biased by a very small number of participants, at least. Stating that "at no point in the article were the findings presented as correlated" is in direct collision with statements from the manuscript, like "a weak negative correlation was found between BMI and oxygen saturation" or "positive correlation was observed between BMI...", etc. Furthermore, it is evident from the graphs that some of the variables analyzed through correlation test are non-normally distributed, for which reason Pearson's test is unsuitable. Describing the results of relative quantification without any statistical assessment is completely unacceptable and misleading. Also, the presentation of results of quantification as bars without any reference to the statistical parameters and the interpretation of error bars is not informative and transparent mode of visualisation. 

Comments on the Quality of English Language

Minor editing is needed.

Author Response

Review Report (Reviewer 2)

The authors have made some corrections, however, the improvements are not sufficient for a meaningful increase in the quality of the manuscript.

Answer: Sorry for this yet. Once again, we improved the manuscript to address all questions and suggestions.  

The corrections are mostly intended to account for major flaws in the presentation of the results (Table orders, missing data...) but the most significant issues with the study design and interpretation persist. Stating these issues as a limitation does not change the fact that the results and their interpretation could be biased by a very small number of participants, at least. Stating that "at no point in the article were the findings presented as correlated" is in direct collision with statements from the manuscript, like "a weak negative correlation was found between BMI and oxygen saturation" or "positive correlation was observed between BMI...", etc.

Answer: We reviewed all the results and discussion text for the correlation tests. We have added two new paragraphs (lines 146-157) to clarify the results of the correlation tests.   

Furthermore, it is evident from the graphs that some of the variables analyzed through correlation test are non-normally distributed, for which reason Pearson's test is unsuitable. Describing the results of relative quantification without any statistical assessment is completely unacceptable and misleading. Also, the presentation of results of quantification as bars without any reference to the statistical parameters and the interpretation of error bars is not informative and transparent mode of visualisation.

Answer: We applied statistical tests in the molecular analysis (relative quantification) and presented the results with average and SEM bars.

Round 3

Reviewer 2 Report

Comments and Suggestions for Authors

The improvements made by the authors are still minor. The statistics is inadequate and the authors neglected my previous comments regarding the distribution of data and the usage of pearson's correlation test. They should either use a non-parametric test or transform the data (log). Furthermore, they tested the results on gene expression presented as fold-change (2-ddCT) which has non-normal distribution by default with a an unsuitable parametric test. This could substantially influence the results.

Comments on the Quality of English Language

Minor editing is needed.

Author Response

Reviewer’s commentary:

The improvements made by the authors are still minor. The statistics is inadequate and the authors neglected my previous comments regarding the distribution of data and the usage of pearson's correlation test. They should either use a non-parametric test or transform the data (log). Furthermore, they tested the results on gene expression presented as fold-change (2-ddCT) which has non-normal distribution by default with a an unsuitable parametric test. This could substantially influence the results.

Response:

Thank you for your continued valuable feedback on our manuscript. We apologize for any oversight in addressing your previous comments regarding the distribution of our data and the use of Pearson's correlation test. 

To address your concern, we have re-evaluated our correlation analysis methodology. Given the nature of our data, we recognize that the assumption of normality required for Pearson's correlation may not hold. Therefore, we have corrected our approach and conducted the analysis using Spearman's rank correlation test, which is a non-parametric test that does not assume a normal distribution and is more appropriate for our data. We stated clearly in the figure legend as well as the Methods section that the statistical test used for this analysis was not Pearson.

Now regarding the gene expression data, we applied the Mann-Whitney U test, a non-parametric test that does not assume normality of the data distribution.

We believe this correction addresses the concern raised and ensures the validity of our findings. Thank you for bringing this important issue to our attention.

Round 4

Reviewer 2 Report

Comments and Suggestions for Authors

The authors have made corrections that improved the manuscript. Although the Figure corrections were made, descriptive part of the results (line 153) still reffers to Pearson's instead of Spearman's rank coefficient. Text in the lines 154-157 is more appropriate for the Discussion section.

Comments on the Quality of English Language

Minor editing is needed.

Author Response

Dear Reviewer,

Thank you for your observation. We replaced the wrong reference of Pearson at line 153 for Spearman's. We also moved the text form lines 154-157 into the discussion (line 281), as you suggested. Hopefully we had attended to all your suggestions to improve the quality of our manuscript.